# Re-Learning EXP3 Multi-Armed Bandit Algorithm for Enhancing the Massive IoT-LoRaWAN Network Performance

**DOI:** 10.3390/s22041603

**Published:** 2022-02-18

**Authors:** Samar Adel Almarzoqi, Ahmed Yahya, Zaki Matar, Ibrahim Gomaa

**Affiliations:** 1Department of Electrical Engineering, Faculty of Engineering, Al-Azhar University, Cairo 11651, Egypt; dr.ahmed.yahya@azhar.edu.eg (A.Y.); dr.zakimattar@azhar.edu.eg (Z.M.); 2Computers & Systems Department, National Telecommunication Institute (NTI-Egypt), Ministry of Communications and Information Technology, Cairo 11112, Egypt; igomaa@nti.sci.eg

**Keywords:** IoT, LoRaWAN, LPWAN, MAB, wireless node

## Abstract

Long-Range Wide Area Network (LoRaWAN) is an open-source protocol for the standard Internet of Things (IoT) Low Power Wide Area Network (LPWAN). This work’s focal point is the LoRa Multi-Armed Bandit decentralized decision-making solution. The contribution of this paper is to study the effect of the re-learning EXP3 Multi-Armed Bandit (MAB) algorithm with previous experts’ advice on the LoRaWAN network performance. LoRa smart node has a self-managed EXP3 algorithm for choosing and updating the transmission parameters based on its observation. The best parameter choice needs previously associated distribution advice (expert) before updating different choices for confidence. The paper proposes a new approach to study the effects of combined expert distribution for each transmission parameter on the LoRaWAN network performance. The successful transmission of the packet with optimized power consumption is the pivot of this paper. The validation of the simulation result has proven that combined expert distribution improves LoRaWAN network’s performance in terms of data throughput and power consumption.

## 1. Introduction

Nowadays, the LoRaWAN system can be considered a primary key of IoT services and applications. Long Range (LoRa) targets deployments where nodes have limited energy supply (battery powered). The long-range and low-power nature of LoRa makes it an interesting candidate for smart sensing technology in the civil infrastructures of most IoT applications [1].

LoRa technology uses Chirp Spread Spectrum (CSS) modulation that consumes lower power than other modulation technologies. The chip signal varies its frequency linearly with time within the available bandwidth. Moreover, that makes the LoRa signals resistant to noise, fading, and interference. The number of data bits modulated depending on the parameter Spreading Factor (SF). LoRa uses six orthogonal SF in the range of 7 to 12, which provide different Data Rates (DRs), resulting in better spectral efficiency and an increased network capacity. LoRa physical layer technology was introduced by Semtech. It also has two other parameters; bandwidth (BW) can be set to 125 kHz, 250 kHz, and 500 kHz m and it uses forward error correction, adding a small overhead to the transmitted message, which provides recovery features against bit corruption. It is implemented through a different Code Rate (CR) from 4/5 to 4/8 (denoted CR = 1 to CR = 4, respectively) [2].

LoRaWAN is open-source connectivity introduced by the LoRa alliance [3]. It is a layer two protocol that responds to control the node modulation parameters setup, security, channel access, and energy saving functionalities. LoRaWAN has Class A and is mandatory in all LoRa node channel access strategies; it is designed to be the most energy-efficient mode. Class A optimizes the node energy by controlling the down-link receive windows (RW) that keep the LoRa node in sleep mode as much as possible. In Class A, after sending messages, the nodes expect an ACK from the network server during two pre-agreed time-slots known as Receive Windows (RWs), which use ALOHA random multiple access protocols [3,4]. The ALOHA allows nodes to transmit as soon as they wake up and exponentially back off for saving power as much as possible and use low signaling overhead as possible. Moreover, it uses light encryption and authentication mechanisms that can be configured during activation.

The LoRaWAN network uses star-of-star network topology with single-hop as shown in Figure 1 to keep network complexity as simple as possible and maximize energy saving. It has simplicity in the configuration in addition to the firmware updates that can be sent over the air [5,6]. That makes LoRaWAN efficient in terms of the deployment cost.

In addition, The traditional LoRaWAN protocol runs a simple control mechanism to coordinate the medium and nodes through commands. The commands are identified by an octet identifier called command identifier (CID)); the commands are processed in the network server [3]. Usually, the nodes do one simple specific task to minimize energy consumption. The open-source LoRaWAN protocol aims to improve and solve all medium and network congestion issues that face the massive IoT network performance [7].

Most IoT applications consist of a massive number of nodes. All these nodes communicate with the GW for transmitting the collected data or signaling for determining the communication channel. The LoRa has variance transmission parameters available that can be used for transmitting such as BW, SF, Transmission Power (TP), Channel Frequency (CF), and the Coding Rate (CR). However, the huge amount of communication over the GW can cause:-Dropping down the GW.-Delaying the data transmission.-Higher incorrect data received.-Extra Consumption of energy.

Most of the IoT remote nodes had been built based on isolated batteries. Also, They have to be working for five to ten years as healthy nodes. Indeed, the LoRaWAN consumes more power due to unavoidable circumstances such as re-transmissions, caused by link impairments [8]. So, choosing an appropriate transmission parameter to compromise between battery consumption and frequent packet loss is a challenge for the LoRaWAN configuration. There are several works that either evaluate the performance of LoRa nodes or reserve the derivation of transmission policies. Our approach works on optimization of the LoRa node transmission performance by deriving transmission policies that optimize both performance and power consumption.This approach focuses on improving the performance of the IoT-LoRaWAN networks in the adversarial environment

The rest of this paper is structured as follows. Section 2 highlights related work and explains the EXP3 adversarial MAB algorithm. Section 3 discusses the problem and introduces proposed approaches for the LoRa smart node. Section 4 outlines the contribution of this approach. Section 5 discusses the proposed approach and its implementation. Section 6 outlines the simulation results and performance evaluation. Section 7 concludes this research paper and recommends its future extension directions.

## 2. Related Works

Finding an optimum configuration for improving IoT-LoRaWAN network performance is a challenge. LoRaWAN protocol has attracted the attention of the academic and industrial community, which led them to study different aspects of LoRa protocol such as coverage, interference, link quality, and other performance improvements. So, the related works section is separated into two sections: Section 2.1: Literature review and Section 2.2: An adversarial MAB EXP3 algorithm.

### 2.1. Literature Review

The IoT design objectives [9] are energy efficiency, scalability, and highly dynamics with a flexible network since the number of nodes can be very high, reliable, robust, and self-healing. Handling of such systems is feasible if the network configuration is automated and adaptable for the actual situations. LoRa technology is designed to achieve the goals of IoT objectives more than various techniques.

The LoRa CSS modulation that enables LoRa to be resistant to interference is illustrated in [10] with respect to the gain of this modulation. In [11] the interference in the present channel orthogonal within different SF values using the test-bed experiment is studied. Researchers performed LoRa coverage in different regions from the demodulation point of view [12].

Reference [13] summarizes LoRa application scenarios with open search areas and highlights the bottlenecks; studies exist to overview the new technology. One of the recently developed solutions for LoRa is ambient energy harvesting. This new technique has back-scatter signals for transmission. The back-scatter signals’ transmission used the existing radio frequency signals such as radio, television, and mobile telephony to transmit data without a battery or power grid connection. That enabled battery-free LoRa devices. Moreover, it enabled wireless power charging solutions for wireless end-nodes.

Adaptive Data Rate (ADR) is a mechanism in LoRaWAN networks to control coverage, interference, and energy consumption. The ADR reduces frame transmission time since nodes closer to BSs use lower SF values and have higher transmission rates, which will minimize channel usage and energy consumption. The ADR currently implemented in LoRa nodes that are designed to optimize the data rates and the transmission interference of the end-nodes takes into consideration the network condition [13]. The ADR algorithm is based on the Signal to Interference and Noise Ratio (SINR) of the last 20 transmissions. The ADR feature should be enabled whenever an end-node has a sufficiently stable radio channel [14]. The ADR is not able to work in a changeable environment. It does not define an algorithm to control node transmission rates. Moreover, it is centralized decision-making.

This centralized architecture is not a practical solution for the IoT system that increases by billions every year. References [15,16] aimed to study lightweight learning methods to optimize the performance of the LoRaWAN technology. These studies compared nodes using multi-armed bandit (MAB) as a lightweight learning method for a decentralized optimization of the channel choice. The simulations compare the standard ADR mechanism with the Upper-Confidence Bound (UCB) and Thompson Sampling (TS) optimization MAB algorithms to choose the optimal SF. That illustrates that the MAB algorithms are much better than an ADR mechanism in the trade-off between energy consumption and packets loss. Moreover, the decentralized decision-making building on the lightweight multi-armed bandit (MAB) learning algorithms led to making each node have sequential decision making under uncertainty. This allows the trade-off between exploration and exploitation. In LoRa, smart nodes increase the LoRa link quality as well as the LoRaWAN network performance for reducing the re-transmission and signaling. The smart node operations can be classified into (i) micro-controller operations and (ii) wireless transmissions. The wireless transmission consumes more power than micro-controller operations [17,18].

Article [19] evaluates the performance of UCB and TS MAB algorithms in IoT networks. The IoT network has two types of devices: Static devices that use only one channel (fixed in time) and dynamic that select a transmission parameter each time. The TS algorithm outperformed the UCB algorithm in fitting the IoT network. While the dynamic devices are below 50% or higher than 50% both algorithms almost have the same performance. However the UCB algorithm fits into the massive IoT networks, it is not suitable for the non-stationary and non-Identical and Independent distribution (IID) environment. The EXP3 is an IID and adversarial non-stationary environment MAB algorithm. Articles [20,21,22] implement nodes by using the EXP3 optimization algorithm in the LoRaWAN network and evaluate its performance.

Reference [23] evaluates the performance of LoRaWAN by using the EXP3 adversarial MAB optimization algorithm in the LoRa network configured with 100 nodes to demonstrate the usefulness of physical phenomena in LoRaWAN such as the capture effect on the inter-spreading factor interference.

### 2.2. An Adversarial MAB EXP3 Algorithm

The EXP3 in [23] is used for reducing the signaling and re-transmission between a node and the GW because the transmission between the node and the GW consumes power more than the calculation accord locally in the node, especially with the simple learning algorithms MAB. This is built on choosing the parameters that increase the sum of reward (ACK) as possible (the successfully packet received). So, the gain is defined as the summation of the rewards for each action (transmission parameter). Then the beast parameter is chosen that has the maximum ratio between the gain and the probability of transmission for each parameter (decisions have to be taken over time or discrete turns). After that, updating the weight $w_{i}\left(t\right)$ of the actions that have the maximum profit (gain) value from the profit set (*g*). The weight of each expert is updated; the update procedure takes as input the best (profit) gain and numbers of actions and the algorithm learning factor switching rate γ as in the below explanation. The algorithm expects the beast arm that has the beast weight indicator. As follows for each time step t=1,2,…,T:

(1) The learner selects an arm with random initialization, also known as an action or forecaster $I_{t}\in \left[K\right]=\left\{1,2,\ldots ,K\right\}$.

(2) The environment (adversary or opponent) receive a gain vector $g\left(t\right)=(g_{1}\left(t\right),\ldots ,g_{K}\left(t\right)\in \left[0,1\right],$ where $g_{K}\left(t\right)$ is the gain (reward) associated with arm i∈[K].

(3) Simultaneously, the learner sees the maximum gain $g_{I_{t}}\left(t\right)$, while ignoring the gains of the other arms. Is it still possible in this situation to avoid losing data?

The learner’s goal is actions chosen to accumulate as much gain as possible during the horizon to attain regret bounds in the high probability or expectation of any possible randomization in the learner’s or environment’s methods.

The learner performance measure is the source of regret. It is the difference between the overall gain of the best decision and what is predicted (learner picks):(1)$$R\left(T\right)=\sum_{t=1}^{T}g_{I_{t}}\left(t\right)-max_{i\in [K]}\sum_{t=1}^{T}g_{i}\left(t\right)$$

The environment is oblivious if it selects a random sequence of the action set irrespective of past actions taken by the learner from the gain vector, and it will be non-oblivious if it is allowed to choose an adaptive selection g(t) as a function of the past actions $\left\{I_{t-1},\ldots ,I_{1}\right\}$ [20].

The expectation integrates over the learner-injected randomness to prove constraints on the real regret that hold with high probability, which is considered a significantly more difficult task that can be performed by making considerable adjustments to the learning algorithms and doing much more complicated analysis. That means the key challenge is constructing reliable estimates of the gains $g_{i}\left(t\right)$ for all i∈[K] based on the single observation $g_{I_{t}}\left(t\right)$.

The EXP3 is the most widely used algorithm for non-stationary, non-identical, and independent distributions (IID), in which the gain (rewards) is chosen by an opponent. As a conventional online learning algorithm, it can develop an exponentially weighted forecaster model. It generates an arm $I_{t}=i$ with a probability proportional to $w_{i}\left(t\right)$ for all *i* and the algorithm learning factor or switching rate η = γ parameter.
(2)$$P_{i}\left(t\right)=\left(1-\gamma \right)\frac{w_{i}\left(t\right)}{W_{t}}+\frac{\gamma }{K}$$
where
(3)$$W_{t}=\sum_{j=1}^{K}w_{j}\left(t\right)$$

Update the weights of the actions:(4)$$w_{i}\left(t+1\right)=w_{i}\left(t\right)exp\left(\frac{\gamma }{K}g{*}_{j}\left(t\right)\right)$$

However, it achieves an optimal regret in a non-stationary environment that chooses random oblivious action before the beginning of the transmission without iteration action switch [21].

#### EXP3 Limitations

The decision-making problem appears in facing partial information (adversarial environment) due to the collision, where decisions have to be taken over time or discrete turns and impact both the rewards and the information withdrawn. The objective is to maximize the accumulated reward or equivalently minimize the accumulated cost over time. The EXP3 algorithm achieves an optimal regret against an oblivious opponent who guesses rewards before the beginning of the game, with respect to the best policy that pulls the same arm over the totality of the game. EXP3 is not built for arm switch; it achieves a high regret. The algorithm does not explore the drift [22]. The runs where the drift is detected obtain a low regret and the runs where the drift is unseen obtain a high regret. Moreover, the convergence times for the EXP3 algorithm are long, in the order of 200 kh [23].

## 3. Problem Statement

The IoT remote nodes have generally limited energy resources. The aim is to minimize the energy consumption and the packet losses of each node in the IoT-LoRaWAN networks. However, the EXP3 optimization algorithm is dependent on the environment; it has a long convergence time, in the order of 200 kh. As in the previous section the EXP3 is built to respect the best policy that uses the same parameter over the totality of the transmission, which changed by receiving the ACK or the maximum number of the re-transmissions (seven times). The algorithm does not detect the best arm changes during the re-transmission that obtain inefficient energy consumption. These weakness are overcome by our modification M-EXP3 which focuses on increasing the network performance in the long convergence times. The convergence time will be significantly affected by the algorithm selection pattern at each transmission; the modified M-EXP3 achieves controlled regret with respect to policies that allow node switches the parameters during the run (re-transmission) and in addition allows play *N* different arms during the run and shows a regret bound (non-oblivious).

## 4. Work Contribution

This research paper is interested in LoRaWAN decentralized decision making with optimizing transmission parameter selection. The solution decreases the reasons for re-transmission by improving the LoRa communication channel quality. However, the available transmission parameters and the selection methodology for choosing the best transmission parameter affect the channel quality.

The current work proposes a Modified-EXP3 (M-EXP3); this approach modifies the EXP3 smart node agent to take expert advice in the calculations of the parameter choice probability distribution for improving the LoRaWAN performance. The M-EXP3 algorithm with expert advice modification for transmission parameter optimization is seen in Algorithm 1 and Figure 2.

The effects of this modification M-EXP3 are compared with experiment 1 EXP3 in [23]. The smart node in [23] has an agent that chooses the best transmission parameters for packet *j* (action $a_{j}\left(i\right)$) to send its data with minimum regret bounds. As explained before, the regret is the difference between the cumulative rewards of the picker and the one that could have been learned by a policy assumed to be optimal.

The difference between the total reward of the algorithm (expected) and the total reward of the best choice $r_{j}\left(i\right)$ is shown in Figure 2. The best choice keeps the regret to a minimum.

The EXP3 algorithm [22,23] is based on exponential importance sampling, which attempts to be an efficient learner by placing more weight on good arms and less weight on ones that are not as promising. As each new packet has to be transmitted, the optimal parameter may be different from the optimal parameter at the previous one. The algorithm detects when the best arm changes.

## 5. Proposed Approach and Implementation

In the following two sections we will explain in more details the modified algorithm and its implementation.

### 5.1. The Modified EXP3 (M-EXP3)

The proposed approach for smart node agent M-EXP3 is used to take expert advice in the calculations of the parameter choice probability distribution that works to improve the LoRaWAN performance. The M-EXP3 algorithm with expert advice modification for transmission parameter optimization is seen in Algorithm 1 and Table 1 that introduces the proposed approach parameter’s description. The M-EXP3 is the regret against arbitrary strategies. It allows to play *N* different parameters during the re-transmission for detecting the changeable in the best parameter and allows arm switches during the run. It uses a regularization method on the reward estimators to ease parameter selection. The algorithm ranks all sequences of actions according to their “hardness”; saved for each action as an expert distribution $B_{i}\left(t\right)$ over the weight of the action $a_{i}\left(t\right)$ at time *t* with an expert (advice) $B_{i}$ is a sequence expected regret for any sequence of pulls (trams mission). At each turn a proportion of the mean gain achieves controlled regret with respect to policies that allow arm switches during the transmission; M-EXP3 is able to automatically trade off between the return profit of a sequence j and its hardness Bi the result from playing N different arms during the run. Hardness-tuned parameters α and γ are the regret against EXP3 arbitrary tactics. The discount factor α of M-EXP3 hinders the convergence leading to a higher regret. M-EXP3 has a discount factor that achieves an active strategy at each turn concerning a proportion of the mean gain and achieves controlled regret that allows arm switches during the run. The best arm uses an unbiased estimation of the cumulative reward at time *t* for computing the choice probabilities of each action, then rearranges the actions in the hardness $B_{i}\left(t\right)$. The weight of each expert is updated; the weight update procedure of the M-EXP3 takes as input the best (profit) gain and numbers of actions and the algorithm learning factor switching rate η = α∗γ
α that decreases the drift detection. The upper confidence bound of action is $I_{t}$ which has the highest probability $p_{K}\left(t\right)$ (see Algorithm 1); if it has a smaller value lower than the confidence bound of another action $i_{t}$ on the present interval *t*, the detector makes a detection drift.
**Algorithm 1** The M-EXP3 Algorithm with the Modification.Parameters: η=γ∗α in [0,1] where α>0 is a discount factor
initialization: wi(1)=1 for all *i* = 1, …, K.For each time t = 1, 2, …At time t,Receive the experts’ advice vectors BiCalculate, for each action i, the probability
(5)Pi(t)=(1−η)∑j=1Nwi,j(t)Bi(t)Wt+ηKCalculate the sum of the weights of the actions at time *t*:
(6)Wt=∑j=1Kwj(t)Choose action It according to the max distribution Pi(t),Receive a profit for the action *i*:
(7)gi(t)∈[0,1],
(8)g∗i(t)=git(t)/Pit(t) If ACK (rj(i)) is received0 OtherwiseUpdate Bi(t) as the reward (here the reward is a function of the expert in addition to the current action)
(9)yi(t)=Bi(t)·g(t)=∑i=1KBi(t)g∗i(t)Update the weight of each expert
(10)wj(t+1)=wj(t)exp(ηKyi(t))

### 5.2. M-EXP3 Implementation

Figure 3 adds the buffering stage to each node. This buffer is used to save the rank of the action set that calculates per sampling period in case of correct set action. However, an extra sampling period will be added; higher power saving can be obtained due to the improvement of the correct choices through the convergence time. The M-EXP3 has the advantages of dynamic performance reword calculation which offers the possibility of renewing the set actions according to the discount factor limits. Moreover, this work considers both higher power and successful packet reception ratio. However, the system throughput will look relatively low compared to the EXP3, but it has same performance for a longer horizon. The simulation results appear concerning an improvement performance in the convergence time agreement with the proposed modified model. The simulator is a simpy Python realistic LoRa network simulator. However, Python leads all the other languages with more than 60% of machine learning developers using and prioritizing it for development. In addition, the simulation was run in a realistic environment, taking into account the physical phenomena in LoRaWAN such as the capture effect and inter-spreading factor interference. The simulation results show that the proposed simulator provides a flexible and efficient environment to evaluate various network design parameters and self-management solutions as well as verify the effectiveness of the distributed learning algorithms for resource allocation problems in LoRaWAN.

The experiment offers the facility of controlling the LoRa set actions *K* = 6 that is the number of the available transmission parameters set (SFs). The inter dependence between data rate and SF yields in Equation (Equation 11).

The data rate:(11)$$\mathrm{DR}\;=\;\mathrm{SF}\frac{\mathrm{BW}}{2^{\mathrm{SF}}}\mathrm{CR}$$
where, the SF is an integer between 7 and 12, BW is the bandwidth =125 kHz, and CR is the coding rate =4/5. The simulator deals with the received packet according to the sensitivity in Table 2. In LoRa, if a collision occurs between two frames with the same SF and the same frequency, only the LoRa can be the decoded frame with the highest power c, and provided that the power difference exceeds 6 dBm. Moreover, we use the European LoRa characterization shown in Table 2 and Table 3. In addition, packet nodes generate in random distribution with an average sending time of 240 s, simulation time horizon $T=10^{5}$, urban area path loss exponents (*n*) = 2.32 path loss, intercept (*B*) = 128.95, shadow fading (σSF) 7.8 dB outdoor standard deviation, $d_{o}=40$ m, packet length = 50 bytes, cell radius 4500 m, and index time step *t* = 0.1 ms.

The energy consumption per node is equal to Packet emission energy multiplied by the number of transmissions; the energy consumed for one packet is equal to the packet radiation duration (which depends on the SF) multiplied by the transmission power; the number of transmissions represents the number of transmissions to send a successful packet (ACK is received).

## 6. Simulation Results and Performance Analysis

The following figures study the effects of the M-EXP3 transmission parameter selection policy on the performance of the simple LoRaWAN deployment.

The results evaluate the LoRa node energy consumption, successful packet reception ratio, and throughput behaviour through simulation processes shown in Figure 3.

As shown in Figure 4 the results of the modified M-EXP3 can be considered as significant candidates in case of higher power saving as well as the LoRa longer life time. As illustrated, the M-EXP3 improves the node power consumption per successful packet transmitted by 0.02 J, especially in the convergence time.

Figure 5 shows results in a range of a convergence horizon time around 200 kHrs and displays that the EXP3 has a lower successful packet reception ratio compared to the modified M-EXP3. The implemented M-EXP3 depicts the fast response for successful packet reception ratio. It is increased by 12.5% in comparison to the conventional EXP3.

Figure 6 shows the effect of the M-EXP3 on the network traffic from the throughput perspective. It illustrates that the network configured with the M-EXP3 node agent has lower throughput than the network configuration with the EXP3 node agent. The buffering stage to each node affects the system throughput that is relatively low compared to the EXP3; contrariwise, it has a higher successful packet reception ratio that can be highly recommended for the convergence time and the extra longer horizon time. The simulation results are in satisfactory agreement with the proposed modified model.

The modified M-EXP3 improves the optimal LoRa parameters’ choice, which reflects on the LoRaWAN network performance.

## 7. Conclusions

Analysis, modeling, and implementation of the modified M-EXP3 LoRaWAN system have been presented. The M-EXP3 nodes policy that allows changing the parameters during the entire run horizon was implemented for improving the LoRaWAN network performance. However, efficient power consumption and a successful packet reception ratio during the learning period are of course at the expense of the network traffic. Evidently, the M-EXP3 has a discount factor that works on decreasing the regret more than EXP3. The improved M-EXP3 results revealed that the algorithm saves a large amount of energy. They also show the higher success rate of the system in receiving packets. Additionally, a promising throughput profile obtained all that on the long convergence time.

High power saving is obtained due to the improvement of the correct choices through horizon time. The dynamic performance reward calculation of the M-EXP3 offers the possibility of renewing the set actions according to either regression or reward limits. However, the system throughput looks relatively low compared to the EXP3; the results show both higher power and successful packet reception ratio. The simulation results are in satisfactory agreement with the proposed modified model.

The LoRaWAN decentralized decision-making solution with the modified self-management agent (M-EXP3 smart node) improves the IoT-LoRaWAN network performance; that satisfies the requirements of the International Telecommunication Union (ITU) recommendation standard.

Future work will focus on decreasing convergence times issue. The effect of fully decentralized smart nodes will be explored in the future extension of this research work.

## Figures and Tables

**Figure 1 sensors-22-01603-f001:**
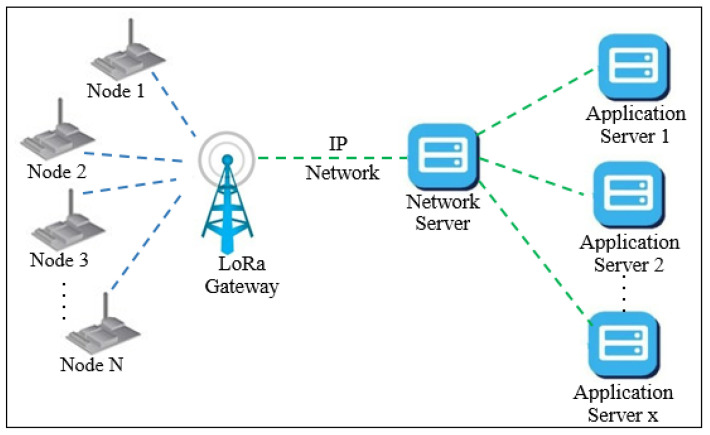
LoRaWAN architecture.

**Figure 2 sensors-22-01603-f002:**
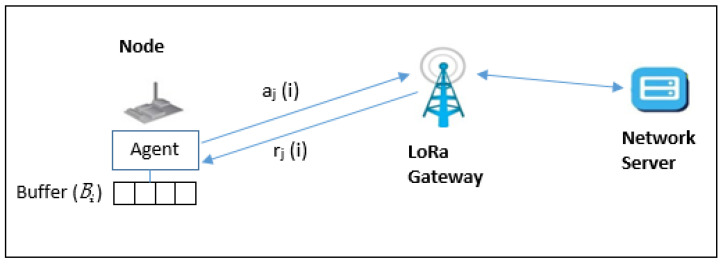
The modified LoRa model.

**Figure 3 sensors-22-01603-f003:**
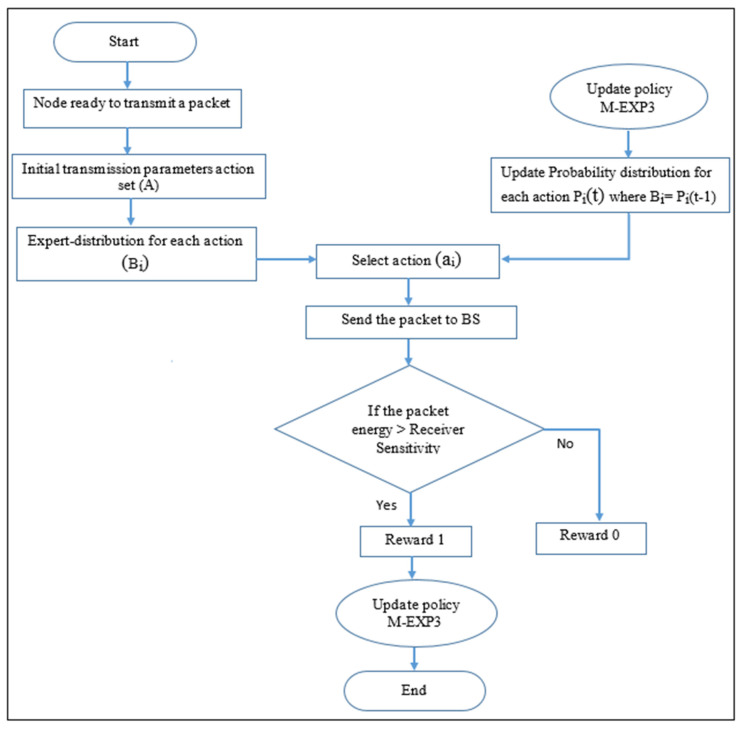
The proposed LoRa smart node simulation flow chart.

**Figure 4 sensors-22-01603-f004:**
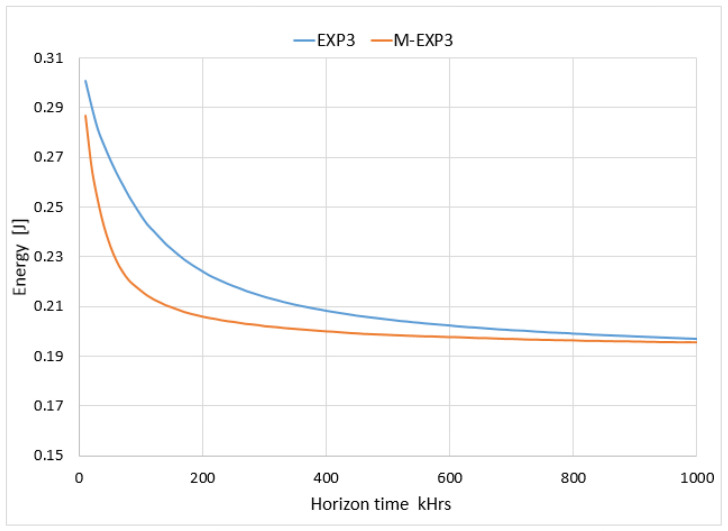
Comparison of the energy consumed per successful transmission packet per node.

**Figure 5 sensors-22-01603-f005:**
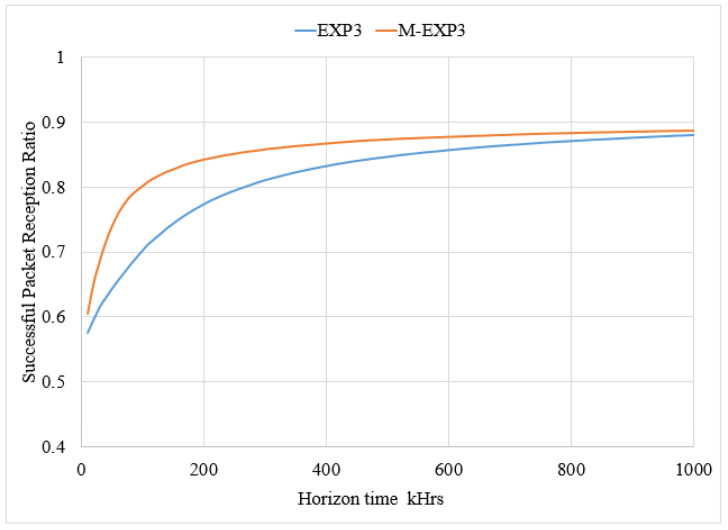
Comparison of the successful packet reception ratio.

**Figure 6 sensors-22-01603-f006:**
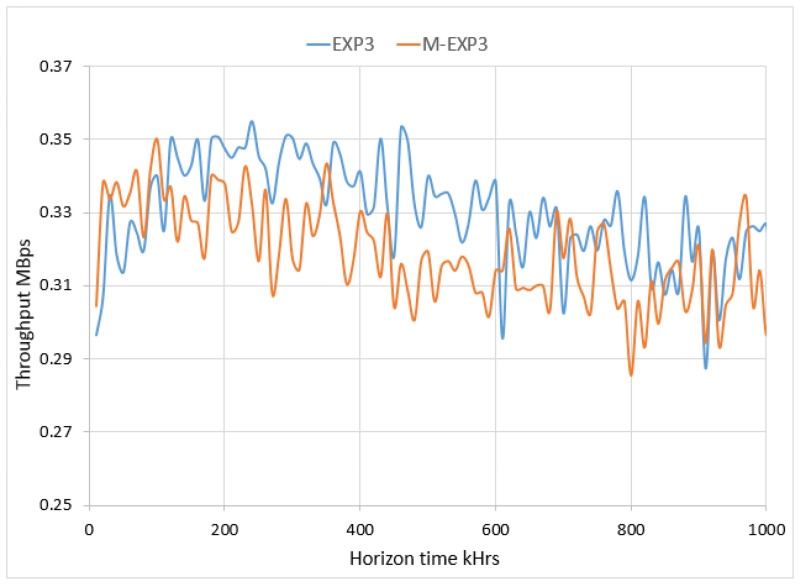
Comparison of the throughput.

**Table 1 sensors-22-01603-t001:** Approach Parameter Description.

Description	Parameters
Index learner policy parameter chooses for a packet	*j*
Index picker parameter chooses	*i*
Reward equal 1 if ACK is received and 0 otherwise	$r_{j}\left(i\right)$
Previous probability distribution as hardness for the actions set	$B_{i}$
Update weight of the action *i* for the next time step	$w_{i}\left(t+1\right)$
Total sum of rewards at each iteration	$g_{i_{t}}$
The profit set for all actions set	*g*
Calculated value of the max profit *g*	$w_{i}\left(t\right)$
Verification step that calculate the gain depends on the previous expert	$y_{i}\left(t\right)$
No. of actions set	*k*

**Table 2 sensors-22-01603-t002:** Europe LoRaWAN parameters.

Parameters	Values
Max. transmission distance	4.5 km
channel	868,100 HZ
Bandwidth	125 kHz
Transmission Power	14 dB
Code rate	4/5
average sending time	240 s
Duty cycle	1%
packet length	50 byte
LoRa Receiver sensitivity	Table 3

**Table 3 sensors-22-01603-t003:** Receiver sens. at BW = 125 kHz.

SF	LoRa Receiver Sensitivity [dBm]
7	−123.0
8	−126.0
9	−129.5
10	−132.0
11	−134.5
12	−137.0

## Data Availability

Not applicable.

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
