# Peer review of "Re-Learning EXP3 Multi-Armed Bandit Algorithm for Enhancing the Massive IoT-LoRaWAN Network Performance"

_sensors, 2022, doi:10.3390/s22041603_

Round 1

Reviewer 1 Report

I have reviewed the paper "Re-learning EXP3 Multi-Armed Bandit Algorithm for Enhancing The Massive IoT-LoRaWAN Network Performance" that presents a modification for the EXP3 algorithm to enhance the performance in terms of throughput and power consumption.

I have the following comments:

1- in figure 4,5,6. the simulation time stopped at 1.3 Khours, Can you extend your simulation time such as reference[20] in order to illustrate the performance of the modified EXP3  after lone time. 

2-could you give a clear explanation for the high performance of your model in the initial time?

3- in figure 6, the throughput has been changed after  1khours, what is the reason? 

4-It is possible to implement this modification on other tools such as OPNET in order to validate your model.

5-in reference [20], said "However, convergence times are long, in the order of 200 hours for the EXP3 algorithm. Therefore, in future work, we need to address the convergence time issue", have you addressed this issue?

6- in Line 170, you mentioned " in addition to time step t=0.1", what is the unit of t.

Author Response

We are uploading (a) our point-by-point response to the comments (below) (the referees' comments),

(b) an updated manuscript with green highlighting indicating changes.

(c) screenshots from the manuscript added to the referees’ comments.

Reviewer 2 Report

The paper presents an alghoritm based on Long Range Wide-Area Network (LoRaWAN) protocol for increasing the performance of IoT applications.

The manuscript should be carefully reviewed, clarifications and additions are necessary.

It is not clear from the manuscript how the energy consumption was calculated.

The units of measurement of the parameters in equations (1)-(9) are missing.

The implementation of the proposed algorithm in the Python computing environment should be provided.

The algorithm is not described with sufficient details to allow another researcher to reproduce the results.

References [1] and [2] represent the same paper.

The formatting of the references should follow the Sensors template.

Author Response

(The authors gave the same response as above.)

Reviewer 3 Report

The proposed scheme is a performance enhancement of the LoRaWAN network by communication parameters configuration. Some further discussions are expected to clarify:

  • The introduction section will be better with the problem introduction.
  • The operation and limitation of EXP3 could be introduced in more detail. Will every node calculate and configure the parameters or the GW will do for all nodes? Will the proposed scheme configure the contention window in collision control protocol? 
  • Configuration parameters (table 1) should be more in detail (i, j, Bi, profit,..) to make fully understand the scheme operation.
  • Some paragraphs are very short. Should consider to improve them.
  • How does the energy information in Figure 3 connect to the proposed scheme? Does M-EXP3 control the transmission power?

Author Response

(The authors gave the same response as above.)

Round 2

Reviewer 2 Report

The authors took note of the reviewers' comments and improved the presentation of their manuscript accordingly.

Reviewer 3 Report

Thanks for your contribution.